# Experiences of Hearing Parents of Children with Hearing Loss: A Qualitative Study

**DOI:** 10.3390/children10071129

**Published:** 2023-06-29

**Authors:** Gül Dikeç, Eda Türk, Esin Yüksel, Kübra Çelebi, Meltem Özdemir

**Affiliations:** Department of Nursing, Faculty of Health Sciences, Fenerbahce University, 34758 Istanbul, Turkey; eda.turk@stu.fbu.edu.tr (E.T.); esin.yuksel@stu.fbu.edu.tr (E.Y.); kubra.celebi752@gmail.com (K.Ç.); meltem.ozdemir@stu.fbu.edu.tr (M.Ö.)

**Keywords:** child, hearing loss, parents, qualitative research

## Abstract

This qualitative study was carried out with a phenomenological design to determine the life experiences of the parents of children with hearing loss. The study sample consisted of twenty parents, who did not have hearing loss, of children with hearing loss registered in a special education and rehabilitation center. The data were collected through in-depth interviews in Istanbul between December and January 2022. Colaizzi’s phenomenological interpretation method was used for qualitative data analysis. It was determined that parents of children with hearing loss experienced anxiety, sadness, and happiness during diagnosis. They needed more information at first, but then they gained knowledge in the process, and it took work to accept this process. They stated that having a child with hearing loss requires more time, responsibility, and effort than other children. This situation affects their social life, and they experience interpersonal conflicts. When parents were asked how they coped, they said they did it through prayer, social support, or time to themselves. It can be recommended to apply psychosocial intervention programs to the parents of children with hearing loss, especially their mothers, from the first diagnosis process.

## 1. Introduction

Hearing loss is defined as the inability of one or both ears to fulfill their hearing ability. It is also diagnosed when an individual cannot hear a sound in any range. Hearing loss is examined widely, from mild to severe degrees [1]. The Ministry of National Education Special Education Services Regulation in Turkey (p. 2) defines an individual with hearing loss as “an individual who needs special education and support services due to partial or complete loss of hearing sensitivity” [2].

Hearing loss is recognized as one of the most common types of disabilities worldwide and one of the most common congenital and acquired diseases in children. While 3–4 out of every 1000 children have unilateral or bilateral hearing loss [3], according to Turkish Statistical Institute (TSI) data, the rate of children aged 2–14 years with hearing difficulties in our country is 2% [4]. Accordingly, the World Health Organization recommends that newborn hearing should be tested within the first three months of life, and newborn hearing screening tests have been routinely performed by the Ministry of Health in Turkey since 2004 [5]. With the widespread use of newborn tests in Turkey, hearing loss diagnosis and treatment processes have accelerated [6].

Early diagnosis of childhood hearing loss is crucial, as it affects language skill acquisition, interpersonal relationships, and academic and social life [3,7,8]. Accordingly, families should be informed about the diagnosis, treatment, equipment, and rehabilitation of each hearing impairment detected by experts [1]. In Turkey, hearing loss in the family is often reported to parents by otolaryngologists. Families experience anxiety and fear when they first hear about the disability, and specialists may not know how to communicate this situation to parents [1]. Therefore, explaining the initial diagnosis to parents and specialists and providing appropriate special education and rehabilitation is also essential.

In Turkey, auditory rehabilitation practices for children often include auditory and verbal therapy [6]. Training is conducted through two-hour individual and one-hour weekly group lessons [2]. The education program aims to help children gain language, hearing, and social skills. Auditory and verbal therapy includes interventions that hearing parents can quickly implement based on daily routines. These interventions help children with hearing loss to use all sensory areas. Families of children with hearing loss need to incorporate these interventions into their lives [6].

On the other hand, auditory and verbal therapy rehabilitation studies cannot be used for deaf mothers and fathers who use sign language. For this reason, the “total communication” approach has commonly been used for these children. Total communication aims at using communication skills in the whole sensory area. It is used in children with a late diagnosis or other disabilities. Sign language and senses such as smell and touch are also used in total communication. Communication is enriched by the use of gestures and mimicking in sign language. However, sign language is spreading more slowly, as individuals with hearing loss often grow up in hearing families [6].

The initiation of special education and rehabilitation processes and medical follow-up of a child with hearing loss affect the daily routines of both the child and the family [7]. Parents who experience negative emotions when they learn about the initial diagnosis may not know how to cope with devices, technology, and special education [3,7]. Responsibilities increase due to the treatment and rehabilitation of their children, and parents feel the need to spend much time with and protect their children with hearing loss. Having a child with hearing loss may cause changes in parents’ social lives [1,7] and may be a source of stress for parents. In addition, increased economic expenditure due to the need for devices, batteries, training, rehabilitation, or inadequate perception of social support may increase the risk of stress, depression, anxiety, and loneliness in parents [7,9]. 

Studies have often examined parents’ subjective experiences or educational needs of children with other disabilities, especially autism or other developmental disorders. However, no qualitative studies in the literature examine parents’ life experiences with children with hearing loss in Turkey. From this point of view, this study aimed to determine the experiences, feelings, thoughts, and experiences of parents with children with hearing loss. The findings of this study will shed light on the literature in the preparation of mental health programs that can be created to increase the psychological resilience of parents with children with hearing loss.

## 2. Materials and Methods

In the reporting of this study, the Consolidated Criteria for Reporting Qualitative Studies (COREQ) (Appendix A) criteria [10], which are frequently used in reporting qualitative studies, were followed. 

### 2.1. Aim and Study Design

Since this study aimed to determine the life experiences of parents of children with hearing loss, it was conducted using qualitative research with a phenomenological design. Phenomenological studies create a suitable research ground for the events, experiences, perceptions, orientations, concepts, and situations experienced by individuals [11]. The research sought to answer the following essential question; “What do parents with children with hearing loss experience daily?”.

### 2.2. Settings

The study data were collected in February–March 2022 in a special education and rehabilitation center on the European side of Istanbul, serving only children with hearing loss.

### 2.3. Participants

The purposive random sampling method, which is frequently used in qualitative research methods, was used [12]. The study population consisted of all the parents of children with hearing loss enrolled in the special education and rehabilitation center. The sample of the study consisted of Turkish-speaking and literate parents who had a child with hearing loss aged 0–6 years old and registered in a special education and rehabilitation center, used cochlear implants or hearing aids, and who did not have hearing loss themselves. There was no time restriction for the first diagnosis of the children. In contrast, parents who did not have hearing loss, a family member, or another child with hearing loss participated in the study. Due to the lack of an exact sample size for qualitative research, the study data collection was terminated when the study data reached saturation, in other words, when no new findings were obtained [13]. A total of twenty parents were interviewed. 

### 2.4. Data Collection

The most frequently used data collection method in qualitative research is in-depth interviews. In interviews, individuals can express experiences and meanings they were unaware of or thought about [11]. In this study, a semi-structured interview form was used. This form consisted of two parts. In the first part, eight questions were about the parents’ age, sex, economic status, marital status, number of children, children with special needs, and employment status [14,15]. In the second part, seven questions were created by reviewing the literature to determine the daily lives and spiritual and social experiences of parents with children with special needs, which was related to the phenomenon (Table 1) [9,16,17,18]. After the semi-structured interview form was created, pilot interviews were conducted with two parents, and the clarity of the questions was tested. The document was revised after the pilot interviews, and the pilot interviews were not included in the findings. Five parents refused the participate in the study. 

In this study, the participants were interviewed three times. In the first interview, necessary information about the study was given, and whether they would participate was determined. The second interview was conducted face-to-face with parents whose children were in individual education classes in a room where privacy was ensured. The interviews lasted an average of 40 min. The interviews were conducted by four senior female nursing students (ET, EY, KÇ, MÖ) who completed psychiatric and mental health nursing courses, received individual therapeutic interview training under the supervision of senior faculty (GD), and had no relationship with the participants. During the interviews, audio recordings were made with the consent of the participants, and the authors also noted the non-verbal expressions of the participants during the interview. The themes and sub-themes were presented to the participants in the third interview, and their approval was obtained. 

### 2.5. Ethical Consideration

Ethics committee permission was obtained from the Fenerbahçe University Non-Interventional Clinical Research Ethics Committee (date of approval 6 February 2023 and protocol code E-88813803-204.01.07-17131), and institutional permission was obtained from the directorate of the special education center. In addition, verbal and written informed consent was obtained from the parents participating in the study. The Declaration of Helsinki and the Law on the Protection of Personal Data were complied with in the study. In using participants’ statements, their anonymity was preserved without giving their names, sex, and age. 

### 2.6. Data Analysis

In this study, the descriptive data of the participants were analyzed in the SPSS 25.0 program. Mean, standard deviation, minimum, maximum, number, and percentage were used to analyze descriptive data. Content analysis was carried out manually; the authors did not use any software. All themes and subthemes were created with the data of the participants. No theoretical framework was used during the analysis. Colaizzi’s phenomenological interpretation method was used for qualitative data analysis [19]. The data analysis steps were as follows: Interviews with the parents were transcribed, and then were read and reread. Important statements about parents’ experiences were selected and analyzed, and codes were created. These codes were then grouped into themes and sub-themes. The results were combined with rich and comprehensive life experiences, and the basic conceptual structure of the phenomenon was defined. 

Qualitative data were analyzed by a psychiatric nurse who received qualitative research training (GD) and other researchers (ET, EY, KÇ, MÖ). Another researcher experienced in qualitative research who was not involved in the study was consulted throughout the process. Five participants were re-interviewed to confirm the themes and sub-themes. After the participants’ approval, no changes were made to the themes and sub-themes.

## 3. Results

### 3.1. Characteristics of Participants

The characteristics of the participants are shown in Table 2. It was determined that most participants were married, women, did not work, and had one child with hearing loss. It was determined that they perceived their economic status to be at a moderate level. 

### 3.2. Content Analysis

Content analysis revealed three main themes and nine sub-themes related to these themes. These central themes were the diagnosis process, its impact on daily life, and coping methods. Under the first theme, “diagnosis process,” the participants expressed the emotions they experienced when they first heard the diagnosis, the stages of acceptance of the process, and the lack of information they experienced. Under the second theme, “impact on daily life”, parents stated that having a child with hearing loss impacted their daily lives significantly since their social lives changed; they took more responsibility in the new process, spent more time with their children, and experienced conflicts. In the third theme, parents expressed how they “coped” and stated that they coped by praying, providing social support, and making time for themselves (Table 3). 

#### 3.2.1. Diagnosis Process

Although all participants mentioned learning about their children’s hearing loss at different times, most participants stated that they experienced many emotions together when they first heard their children had hearing loss. The acceptance phase could be difficult for some and more manageable for others. On the other hand, they said they did not know what to do because the parents had no information about this situation they faced for the first time. 

Emotions

Participants expressed that they were distraught when they first heard their children’s diagnoses; participants were surprised by this situation because there was no individual with hearing loss in their family, and they were concerned about their children and their children’s future. One of the participants described her child’s sadness and the silence that her child experienced as very painful. 

*“Of course, we were distraught. Let me put it this way: Imagine being in a quiet environment. Even in that quiet environment, her silence has a sound. He can hear your heartbeat, and if he steps, he can hear the sound of her walking. So even if he scratches here, he can hear it. However, for a child who cannot hear, even your silence has any sound. So, it is excruciating. Even when he says it, sometimes his voice trembles. That is why it was not easy to adopt him. Now, thank God. I do not do that much now, but it is still excruciating. It is painful when he disconnects from the world completely when he removes the device”*.P3

Some of the participants expressed concerns about the future of their children. In particular, parents stated that they wondered whether their children could speak like their peers, whether they could learn to read and write, and what awaited them in their school life. One participant stated that she was worried about what her child would experience in adult life, wondering whether she would be able to hear her baby crying when her child grew up and became a parent, and that she was upset and worried about this.


*“How did you feel? Of course, I felt sad. Let me tell you my first thought; when she grows up and gets married, how will she hear her baby when her baby cries? That was the first thing that crossed my mind. I do not know why; I was unfortunate then. However, thank God it was something that could be cured. I mean, I am a very good girl...”*
P13

One participant expressed that when she saw other children with hearing loss, she wondered whether her child would speak or not, but when she was first fitted with the device, her child reacted to her smile and laughed, and she was pleased: 

*“We came here and saw the kids here and stuff, but I do not know, is it like this? Will ours be like this, or will he start talking like them or something like that? You cannot put your finger on it. Then he already had surgery. I never forget that moment after the surgery. He had the surgery, and when we put him on his devices, the moment he laughed at my laughter in the car and responded to me was very nice. For example, I will never forget that moment. Because I realized that he laughs because I laugh, that is, he laughs because he hears it”*.P19

Some participants stated that they were concerned about their children’s future and wondered whether they would have a similar life to their other children. One participant summarized this situation as follows: 

*“We learned that he had a severe hearing loss of 90 percent. Of course, we were unfortunate; I mean, it is inexpressible; I mean, as they say, will he ever be able to laugh again? Will he ever be able to return to normal life? While we were thinking, thank God we learned that there is treatment; he is fine now with devices and surgeries”*.P11

Lack of Information

All participants stated that they did not know about devices, implants, surgery, hearing loss, and the education process before their child was diagnosed, that they had never experienced a similar situation, and that they learned by experience. One participant summarized that she realized her child’s hearing loss late, that she later made a mistake in choosing a rehabilitation center due to a lack of information, that the planned surgeries were canceled due to the pandemic at that time, and that this situation reflected on them as a waste of time: 

*“We noticed A.’s hearing late. I mean, we noticed it when A. was about seven months old. Well, when we went there, it was already a pandemic. When the pandemic started, they [surgeons] did not operate immediately. At the age of one and a half, A. underwent surgery. A month later, he was fitted with a cochlear implant. Again, A. did not hear at all in that one month. We started education right after that. Nevertheless, we started in the wrong place. I wasted a year or so. It was empty education. I did not know about this place. A. was educated in a rehabilitation center with children with normal developmental delays. However, A.’s place was here. We did not know anything. That is what they told us. You do not know; it happens to you for the first time”*.P2

One participant stated that she noticed the hearing loss in her child; however, her family prevented her from consulting a doctor because they normalized this situation, and her family’s lack of knowledge was reflected in the process:

*“At that time, we lived with the elders [mother-in-law, father-in-law]. I asked for the test myself. Otherwise, my child would have remained like that. His grandfather, the father-in-law, said, “His father started talking when he was seven.” They were telling me, “He looks like him. Do you know better than doctors?” Then I realized it myself. When I brought him to the doctor, we had to put him on a device. Well, I cannot tell you how I felt then”*.P9

Acceptance

Parents frequently stated that they had difficulty accepting this situation when they heard the first diagnosis, that parents went from doctor to doctor, repeated hearing tests in different institutions and sometimes had to have further tests, and that they could not approve of this situation for their children, and therefore had difficulty accepting it. One participant stated that because their diagnosis process came during the pandemic, many hospitals did not accept them for the operation, and they had problems. Another participant indicated that they were very upset about this situation, that they had no other option but to accept it, and that the process could be worse if they did not accept it: 

*“When he was born, they [health professionals] said there might be water [amniotic fluid], etc. They subjected us to hearing tests for about three months, but he did not pass. At one point, he passed one of them and failed the other. These were tests for normal hearing problems. Then they said let three months pass, and they were not sure. So, they took the Bera test to have a definite result. It is the test that gives the most accurate result. When it was done there, there was a high hearing loss. It was 95 decibels. Of course, it makes me sad because nobody wants anything to happen to their child. Yes, it is difficult to accept it at first, but you realize that if you do not accept it, the outcome will be worse. We did whatever it took”*.P8

#### 3.2.2. Impact on Daily Life

Most participants reported changes in their daily routines after their children were diagnosed with hearing impairment. Some participants mentioned that they no longer socialize as they used to. Most participants said having a child with hearing loss requires more time, responsibility, and effort. Some mentioned increased conflicts within and between the family. 

Impact on Social Life

Some participants stated that, when they were first diagnosed, they did not go out, confined themselves at home, and isolated themselves from society. One participant explained that although he worked at an audiometry center, he had difficulty coping with this situation and confined himself at home for days:

*“I used to work at a hearing center, and my relative is an audiometrist. You know, something extraordinary happened. “It was a bad thing; we initially had troubled times. I mean, I did not go out or anything. I did not even see anyone for two or three months, then we got used to it, and we continue like this”*.P6

Another participant stated that she used to be more social, but after having a child with hearing loss, she was not as sociable as before:

*“It affected my life. I cannot say it did not. You know, you care for a child, but you have to care for a child with hearing loss more; the responsibility is a lot. I used to travel more before, but not so much now. Thank God I love my children. My older children also helped me a lot. That is how it is”*.P20

One participant stated that they never celebrate Valentine’s Day with their spouse anymore because the day their child was diagnosed coincided with Valentine’s Day:

*“So, on 14 February 2018. What did I feel at that moment? I felt terrible; it was a day when I left the hospital crying. Well, since it was 14 February, it has never celebrated Valentine’s Day until now. It was such a day”*.P19

More time and responsibility

Most of the participants stated that having a child with hearing impairment requires more time, responsibility, and effort than other children. First of all, due to the hearing loss of the children, families feel the need to constantly repeat every direction, spend more time bringing their children to the special education and rehabilitation center, have to keep spare batteries or spare devices with them at all times, and think that they need to be cautious. Some participants said they quit their jobs or did not work because they had to spend most of their time with their children. Participants also said they need to pay more attention to protecting their children with implants from falls and bumps. In particular, participants who have another child without hearing loss stated that they could not allocate equal time to their children and that this situation created anger:

*“It did not affect a lot [life], but it affected it in this way. This time you feel sorry for him a lot; you try to be enough. Of course, you become aggressive. You do not have much time for others [children]. You spend most of your time on him”*.P1

One participant explained that he felt the need to think and back up more when his device broke down abroad or he had to carry a spare battery:

*“How did it affect you? You have to protect the device; this is an A. He does not have a prominent ear, so he uses it as a band. You have to keep a spare battery with you at all times. For example, this summer, I was in Macedonia for three months. Just two days before my arrival, his device broke down. There is nowhere I can meet there. That was not good, for example. It affects you when you cannot hear. He hears, but he gets angry and aggressive because he does not hear clearly. Now that he knows everything, he changes when his battery runs out, “Mom, the battery is dead.” It has those aspects; for example, you must always back up everything”*.P16

One of the participants stated that it was difficult to protect her child against impacts because her child had implants:

*“Your child is not in much trouble. Yes, but he has an implant. You know, being careful is a bit of a thing. The fear of not hitting his head somewhere envelops you so that there is no problem. This situation is not much different from a normal child; only one more device falls off, and it can be tiring to put it on. We can fasten it with a buckle or something; they are more comfortable. The difficult part right now is so that it does not fall off. You understand when you talk; there is no difference from a normal child”*.P8

Conflicts

Under this sub-theme, participants expressed the conflicts they experienced with their spouses, children’s siblings, and other children. The participant who stated that their children were more irritable and angrier than other children and that they had conflicts with their father and siblings added the following:

*“The father is absent until the evening, but the mother is always around, so the mother’s process is more challenging. He clashes with his father; he cannot understand him and gets angry and aggressive because he cannot understand him. It is hard to deal with them; they tease, and so on. I understand and try to understand him; nobody wants to be like this. No one wants to hear it; everyone wants to hear it. This time he attacks his brother or something. I say, “I understand you, but there is nothing to do, my son”*.P10

One participant stated that she could not get support from her husband and had conflicts, that she changed due to economic difficulties, but that she ignored all these things in this process and that what she went through made her more resilient:

*“After a while, I stopped seeing my husband. I left him behind and worried about my child. I thought the child’s troubles were more important than everything and focused on the child. Everything was one problem after another. Financial issues, my husband, and my other child are all on top of each other. As they say, a problem does not destroy a person, it strengthens him/her, and it did”*.P4

#### 3.2.3. Coping

Under this main theme, participants described how they coped when their children were first diagnosed, afterward, or in the face of difficulties. Frequently, participants said they coped by praying and giving thanks, utilizing social support, and making time for themselves.

Coping through Prayer

Most participants stated that they cope with this situation by referring it to God, praying, saying “God is great”, taking refuge in Him, being patient, and being thankful that this problem can be solved with a device or surgery. One participant stated that they rebelled at the beginning, but as they saw the disabled people who they thought were in a worse situation, they were grateful for this situation: 

*“I mean, thank God, there are worse things again. At first, I rebelled, God forbid, of course, a little bit, but then I realized that there are worse ones. Thank God there is a remedy, of course. Of course, if a child is blind or has a crippled leg, they can continue their normal life now. Thank God, this is the experience”*.P18

Social Support

Most participants stated that they received support from their spouses or families during this period and coped with this process thanks to social support. One participant expressed the help of her spouse as follows:

*“My husband is my constant supporter. We came together; we mostly attended their training together. We do everything together, so my husband’s family is also my supporter. I also take my child to other institutions. Right now, he speaks about four words “Mom, Dad.” He is like children of the age he should be”*.P11

Another participant stated that she did not receive social support from her husband and family, that only certain people were supportive, and that they were interested in why or from whom the disability originated in this process: 

*“We did not have any problems with my husband. I always cared for him, taking him to classes, etc. My husband was working; he was not very interested. I used to go to the hospital alone. I have a sister. She took care of me; she was always with me at that time. My husband’s family did not do much; they were not interested. In fact, you know what happens in families; who did it [the gen causing the hearing loss] come from, us? Was it from you? Why did this child become like this [hearing loss]? Questions were asked. That is why we did not see each other for a while. My brother helped me more”*.P6

Making Time for Yourself

Some participants stated they could cope by making time for themselves, going out with friends, caring for their children, or doing their favorite activities. One participant said that she coped with this situation by shopping or doing her favorite activities: 

*“Since A. is with me 24/7, we do everything together. Since A. has a feeding problem, I can only trust and leave A. with my sister. She is the only one who can feed him around. I usually go out with my friends. With them, you know, they are working. I cannot work; I am at home. The more I hear about their work, the more I do things. My mind wanders; I can only see my friends to distract myself. I go shopping; the hairdresser is my favorite place; it is nice”*.P16

This study aimed to determine the life experiences of parents of children with hearing loss, and the data obtained were discussed in line with the literature on the main themes of diagnosis, impact on daily life, and coping. Families expressed complex emotions when they learned their children had hearing loss, including sadness, confusion, anxiety, and happiness after their children were treated. This happiness experienced by the families may be related to the fact that the child responded to the treatment or that they thought it was a sign of improvement. In the literature, it has been reported that families experience anxiety, sadness, loss, and grief when they learn that they have a child with a disability while dreaming of a healthy child [20].

Similarly, parents in this study also experienced the stages of denial and acceptance. Especially at the first diagnosis stage, the behavior of going from doctor to doctor and seeking remedies door to door is associated with denial [21]. In this study, the participants stated that, when they first learned about hearing loss, they went to different institutions and doctors with their children and repeated hearing tests. Later, the participants noted that acceptance was difficult, and they accepted it. Having a child with a disability causes a crisis in the family. In this process, points such as the type of disability, severity of the disability, and response to treatment affect acceptance [21]. In the current study, when families thought about other children with special needs, they expressed that they were lucky to have a remedy for this situation. This result may be related to minimizing disability losses with appropriate treatment, intervention, devices, and rehabilitation. In a study conducted by Deniz et al. [21] with the parents of children with special needs, it was found that the group with the lowest anxiety and the highest life satisfaction was the parents of children with hearing loss. In this direction, the study’s findings are similar to the literature. 

Participants expressed concerns about their children’s future experiences in the present study. Similarly, it has been reported that parents of children with hearing loss or special needs are concerned about their children’s developmental and educational achievements and future experiences [3,20] and the uncertainty about what their children will face when they grow old or die [22,23]. In this direction, the parents’ concerns might be related to their children’s special needs rather than hearing loss, similar to in previous literature. 

In this study, participants stated they lacked knowledge about hearing loss, treatment, and special education after diagnosis. Fitzpatrick et al. [3] also found that parents of children with hearing loss lacked information about the use of devices in the first place. A study conducted in Canada determined that parents needed more information about hearing loss when they first learned the diagnosis [7]. In another study conducted in Turkey, parents suggested that parents should be informed about the legal rights of children with special needs and families, family education about children with special needs, and that this information should be provided regularly [24]. Especially after the denial stage, parents need information about the special education of children with hearing loss, the devices they use, the rights of children and families, and the approach of families to children, and this lack of information should be eliminated. 

Participants in this study reported that having a child with hearing loss requires more time, labor, and responsibility due to coming to special education and rehabilitation. Parents with more than one child and other children who were not deaf expressed that their children with hearing loss were not like the others and that this process was more complicated than the care of other children. Some parents said that they neglected their other children during this process and felt guilty. In the literature, it has been reported that having a child with special needs is associated with spending more time with the child and taking more social responsibility. Similarly, in the study of Turan et al. [23], parents stated that they neglected their other children, spouses, and themselves because of their children with special needs. In this direction, the study’s findings are similar to the literature. 

When the study findings were examined, it was observed that mothers often brought their children to the particular education center and participated in this study. Some participants said they had quit their jobs and were no longer working. Including a child with special needs in life affects the social and professional roles of the mother, as the most crucial burden in care is felt on her shoulders [25]. While the economic expenditures of the parents of children with hearing impairment increase, the departure of one of the parents from work life causes a decrease in family income [26]. In Turkey, the financial support provided by public institutions to parents of children with hearing impairment is limited. Social services provided to children with disabilities vary according to the family’s income level, type of disability, and severity of the disability. However, individuals with hearing loss report that the disability salaries they receive cover their essential needs rather than their medical expenses. Families are charged a fee difference for hearing aids, and families can pay for batteries and spare parts [27]. In addition, in cases of moderate or unilateral hearing loss, families cannot benefit from the allowances due to the low level of disability. Considering both the rehabilitation expenses of these children and the costs brought by hearing technology and the fact that the caregiver does not work, it can be said that families may experience economic losses. In this direction, the importance of the country’s policies to support individuals with disabilities and their families once again emerges. 

Parents stated that their daily routines had changed due to their children’s hearing disability, that they often participated less in social life, and that they no longer celebrated special days. Turan et al. [23] also stated in their study that parents often stayed at home and could not do social activities such as walking or going to funerals or weddings in daily life. Therefore, there is a need for centers that can provide hourly care for children with hearing loss and support their educational achievements. 

Participants reported that their children were more irritable or angry and had conflicts with their spouses, other children, or peers. It is essential to support children with hearing loss in anger management and problem-solving. Psychosocial support programs that include the child and the family should be implemented. The qualitative study by Turan et al. [23] expressed the psychosocial support needs of families and children with special needs. It was determined that families experienced problems between spouses and siblings and that they felt inadequate in managing the behaviors of their children with special needs, such as anger and stubbornness. Accordingly, children with hearing loss and their parents must be supported psychosocially by guidance counselors and school nurses in the schools where they study and by special education and rehabilitation centers. It is vital to support families in managing and coping with this process. 

The current study determined that parents’ coping methods were religious, such as praying, using social support, and making time for themselves. It is thought that using spiritual coping methods and fatalism [22] as family coping methods has culture-specific characteristics. While some parents stated that social support sustained them, others indicated that they did not receive social support in this process. In particular, some participants said that their spouses were unsupportive and ignored the situation. The literature stated that relationship satisfaction and perceived partner support decreased among parents of children with hearing loss [20]. In addition to affecting interpersonal relationships between parents, one participant stated that her spouse’s parents asked her why their child had hearing loss. In these cases, parents may see themselves as the cause of the child’s hearing impairment. They may blame themselves for thinking they have defective genes or did not play a good enough role in diagnosing hearing loss [28]. This situation indicates that parents may be labeled and stigmatized in the family and society due to their children’s hearing impairment. Stigmatization may also increase parents’ stress [20]. Therefore, psychosocial services should be provided in special education and rehabilitation centers with support groups formed by parents of children with hearing impairment. In particular, parents’ expressing their feelings can make them feel that they are not alone. In addition, parents’ lack of knowledge can be addressed in these sessions. Carrying out all these practices in special education centers is vital for parents who cannot leave their children with someone else to spare time for themselves while their children are in individual lessons. Increasing the well-being of parents will also support the well-being of children. 

Limitations

The results of this study were limited to the experiences of the parents of children with hearing loss who participated in the study. Due to the nature of qualitative research, it cannot be generalized. An essential aspect of this study was thoroughly examining parents’ experiences of children with hearing loss in a single special education center. Another limitation of this study was the participation of only one father. While his emotions and experiences were similar to those of the mothers, examining fathers’ experiences deeply in future studies is recommended.

Furthermore, severe hearing loss and other disabilities were not evaluated in the children in the current study. This study contained the themes and subthemes of parents of children both with cochlear implants and hearing aids. No distinction was made in this study. For future studies, it is recommended to evaluate the experiences of parents of children with cochlear implants, hearing aids, and children with disabilities. 

## 4. Conclusions

In this study, it was determined that the parents of children with hearing loss who participated in the study experienced emotions such as anxiety, sadness, and happiness during the diagnosis process, that they lacked information at first. They gained knowledge in the process, and it took work to accept this process. They stated that having a child with hearing loss requires more time, responsibility, and effort than other children, that this situation affects their social lives, and that interpersonal conflicts are experienced. Finally, they said they cope by praying, providing social support, or making time for themselves. Therefore, it may be recommended to implement psychosocial intervention programs for parents of children with hearing loss, especially for their mothers, starting from the first diagnosis process. This intervention should be provided to parents as a routine practice, and every parent should receive this service. Parents should be encouraged to express their experiences and feelings during this process. In addition, their possible need for more information should be addressed, and their questions should be answered. 

## Figures and Tables

**Table 1 children-10-01129-t001:** Open-ended questions in the second part of the semi-structured interview.

1. Can you tell us about yourself? 2. Could you give information about your child? 3. When did you learn that your child has a hearing loss, and how did you feel? 4. How has having a child with hearing loss affected your life? 5. How did having a child with hearing loss affect your relationship with your family and spouse? 6. Did you receive support during this process? (Psychosocial support, etc.) 7. Everyone has areas and times where they have difficulties. How do you cope in difficult times?

**Table 2 children-10-01129-t002:** Characteristics of participants.

Characteristics	(Min–Max) Mean
Age	(23–55) 37.33
Sex	n (%)
Female	19 (95)
Male	1 (5)
Marital Status	
Married	19 (95)
Single	1 (5)
Economic Status	
Moderate	19 (95)
Good	1 (5)
Employment Status	
Employment	3 (15)
Unemployment	17 (85)
Occupation	
Housewife	17 (85)
Cook	1 (5)
Technical Staff	1 (5)
Apartment Staff	1 (5)
Number of Children	
1	3 (15)
2	11 (55)
3	5 (25)
4	1 (5)
Number of children with hearing loss	
1	19 (95)
2	1 (5)

**Table 3 children-10-01129-t003:** Themes and subthemes of the study.

Themes	Subthemes
Diagnosis Process	Emotions
	Lack of information
	Acceptance
Impact on Daily Life	Impact on Social Life
	More time and responsibility
	Conflicts
Coping	Coping through praying
	Social support
	Making time for themselves

## Data Availability

The data presented in this study are available on request from the corresponding author.

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
