# Peer review of "Experiences of Hearing Parents of Children with Hearing Loss: A Qualitative Study"

_children, 2023, doi:10.3390/children10071129_

Round 1
Reviewer 1 Report
Thank you for allowing me to review the manuscript. I think it is very interesting and with a little-addressed perspective, however, it has some aspects to improve by its authors.
Pg. 30 => hearing loss?
Pg. 38 => being a direct quote, it would indicate the page number
Introduction
Since we are talking about deaf children... I think that sign language should be introduced as the natural language of deaf people, and that in a high percentage of countries it is an official language. I also think that there is talk of rehabilitation... interesting frequent intervention strategies and not only verbal therapy... or speech therapy rehabilitation... there is a deaf community... if you want to make a phenomenological approach, you must respect the values ​​and beliefs of this population. I think pages 61-63 should be justified.
64-73 what they describe is general to the field of disability, and there is nothing included about the separation of oralists versus the deaf community.
Pg 91-92 but if the objective is in terms of "the first diagnosis", this question can be misleading
Data collection
Was experience with deaf people determined? Other family members who are deaf? General background of experience.
All the articles focused on asking the interview questions are general from families of people with disabilities, there is none specific to the deaf, nor from their unique perspective with their own language and their own culture...
Pg 123 why were students chosen? And all nursing? How did other types of profiles not be taken? Even if they completed and did the training, wasn't there a researcher with previous experience? Given that they talk about daily life, how could they not have occupational therapy?
Pg. 221 themes/subthemes based on what theoretical framework has this been done?
discussion
In addition to having to reflect more, the limitations of this study are many and not only the participation or not of the parents. They say they use the COREQ_32 but a lot of information is missing to replicate this study
Author Response
|
Reviewer’s Comments |
Authors’ Response |
|
Thank you for allowing me to review the manuscript. I think it is very interesting and with a little-addressed perspective, however, it has some aspects to improve by its authors. Pg. 30 => hearing loss? Pg. 38 => being a direct quote, it would indicate the page number |
Thank you for your contributions.
We added the keywords “hearing loss”.
We added the page number. |
|
Introduction Since we are talking about deaf children... I think that sign language should be introduced as the natural language of deaf people, and that in a high percentage of countries it is an official language. I also think that there is talk of rehabilitation... interesting frequent intervention strategies and not only verbal therapy... or speech therapy rehabilitation... there is a deaf community... if you want to make a phenomenological approach, you must respect the values ​​and beliefs of this population. I think pages 61-63 should be justified. 64-73 what they describe is general to the field of disability, and there is nothing included about the separation of oralists versus the deaf community.
|
Thank you for the clarification. After your comment, we understand that our language is not correct. We used "hearing loss" instead of the "deaf." We also added sign language in the introduction part at lines between 111-119. We did not discuss more. As the study title suggests, we determined the experiences of hearing parents in this study. |
|
Pg 91-92 but if the objective is in terms of "the first diagnosis", this question can be misleading |
We deleted. Thank you for your contributions. |
|
Data collection Was experience with deaf people determined? Other family members who are deaf? General background of experience. All the articles focused on asking the interview questions are general from families of people with disabilities, there is none specific to the deaf, nor from their unique perspective with their own language and their own culture... Pg 123 why were students chosen? And all nursing? How did other types of profiles not be taken? Even if they completed and did the training, wasn't there a researcher with previous experience? Given that they talk about daily life, how could they not have occupational therapy? Pg. 221 themes/subthemes based on what theoretical framework has this been done? |
No, the experiences were limited with the parents who do not have hearing loss. We have described this in the participant's section (lines between 171-173).
Thank your contributions. You are right; there is a gap in this issue.
Since this study was a graduation project of senior nursing students, they collected the data. At all steps, the experienced researcher supervised the students. We added the supervision at line between 209-210. They might not mention this issue because occupational therapy is rare in Turkey.
All themes and subthemes were created with the data of participants. We added this part to the data analysis (lines between 228-229). |
|
Discussion In addition to having to reflect more, the limitations of this study are many and not only the participation or not of the parents. They say they use the COREQ_32 but a lot of information is missing to replicate this study |
We added all limitations that we realized and comments according to Reviewer II (lines between 668-680). Thank you for your contributions. We also checked COREQ, added some points and COREQ checklist as a supplementary file. |
Reviewer 2 Report
Experiences of Hearing Parents of Deaf Children; A Qualitative Study
The topic is of interest as it is common experience how difficult is to communicate a diagnosis of hearing loss to Hearing Parents of Deaf Children. This is a qualitative study based on the answers of 20 participants.
I have a few remarks
Introduction:
Hearing loss is defined as the inability of one or both ears to fulfill their hearing ability fully and is mentioned when an individual cannot hear the sound given in any range, and hearing loss is examined in a wide range from very mild to very severe degrees [1].
The sentence is not so clear. In my opinion it should bee rephrased
Data collection
In the second part, there were seven questions created by reviewing the literature to determine the daily lives, spiritual and social experiences of parents with children with special needs [9, 16, 17, 18].
The seven questions, in my opinion, shoul be listed in this section, explaining how they were choosed.
In this study, the participants were interviewed three times,. I havent’ found, in this section, the presentation of the third one
When did you conduct the first interview with respect to time of diagnosis? Was is it random ?
Results
Table 1 is not clear, it seems that some lines are empty, the caption appears incomplete.
Caption of table 2 appears incomplete too
In my opinion It woul be important to have more information about child‘ age of diagnosis, entity of hearing loss, eventual presence of other disabilities.
It would be interesting to know if the emotional experience of parents of deaf children is similar independently by the use of hearing aids or cochear implants. In other words, do parent’s react differently if they understand that hearing loss is not severe or profound?
Discussion
A limitation of the study is due to partecipation of the study of only one man .
I think this aspect shoud be discussed in this paper. What about males feelings and partecipation ?
Author Response
|
Reviewer’s Comments |
Authors’ Response |
|
The topic is of interest as it is common experience how difficult is to communicate a diagnosis of hearing loss to Hearing Parents of Deaf Children. This is a qualitative study based on the answers of 20 participants. I have a few remark |
Thank you for your contributions. |
|
Introduction: Hearing loss is defined as the inability of one or both ears to fulfill their hearing ability fully and is mentioned when an individual cannot hear the sound given in any range, and hearing loss is examined in a wide range from very mild to very severe degrees [1]. The sentence is not so clear. In my opinion it should bee rephrased
|
We clarified and split the sentence (lines between 32-38). |
|
Data collection In the second part, there were seven questions created by reviewing the literature to determine the daily lives, spiritual and social experiences of parents with children with special needs [9, 16, 17, 18]. The seven questions, in my opinion, shoul be listed in this section, explaining how they were choosed. In this study, the participants were interviewed three times,. I havent’ found, in this section, the presentation of the third one When did you conduct the first interview with respect to time of diagnosis? Was is it random ?
|
We added the questions in Table-1 and explained.
We added third interview as “The themes and sub-themes were presented to the participants in the third interview” between lines 213-214. Thank you for your attention. The interview conducted with parent who have deaf children aged 0-6. There was no restriction about time of diagnosis. We added this information on the participants subheading at lines between 171-172. |
|
Results Table 1 is not clear, it seems that some lines are empty, the caption appears incomplete. Caption of table 2 appears incomplete too In my opinion It would be important to have more information about child‘ age of diagnosis, entity of hearing loss, eventual presence of other disabilities. It would be interesting to know if the emotional experience of parents of deaf children is similar independently by the use of hearing aids or cochear implants. In other words, do parent’s react differently if they understand that hearing loss is not severe or profound? |
We resulted both two tables.
In the diagnostic process, we explained the children's age: "All participants mentioned that they learned their children's in different times." lines 260-261. We did not have any data about severe hearing loss and the eventual presence of other disabilities in children. Therefore, we added these points in the limitations section (lines between 668-680). We did not separate the experiences of parents with children who used cochlear implants or hearing aid. Since they could affect the parents differently, we added this to the limitation section. Thank you for your contributions.
|
|
Discussion A limitation of the study is due to partecipation of the study of only one man . I think this aspect shoud be discussed in this paper. What about males feelings and partecipation ? |
We emphasized this point in the limitation section, too. Thank you for your recommendations. |

Round 2
Reviewer 1 Report
Thanks for the changes, I hope the comments and suggestions from the first review have been useful to the authors. I think they should change gender to sex
Then I put this comment “All the articles focused on asking the interview questions are general from families of people with disabilities, there is none specific to the deaf, nor from their unique perspective with their own language and their own culture...
The authors response is “Thank your contributions. You are right; there is a gap in this issue.”
This needs to be addressed
The discussion, I think a greater debate between children of deaf parents and children of hearing parents is interesting…. CODAs don't even appear in the discussion, and it's very interesting
Author Response
|
Reviewer’s Comments |
Authors’ Response |
|
Thanks for the changes, I hope the comments and suggestions from the first review have been useful to the authors.
|
Thank you for your contribution. We believe that the manuscript was enhanced and improved after your recommendations.
|
|
I think they should change gender to sex
|
We changed the gender as sex. |
|
Then I put this comment “All the articles focused on asking the interview questions are general from families of people with disabilities, there is none specific to the deaf, nor from their unique perspective with their own language and their own culture... The authors response is “Thank your contributions. You are right; there is a gap in this issue.” This needs to be addressed.
|
Thank you for your attention. We had written in the last paragraph (lines 82-85) introduction part before. We have addressed. |
|
The discussion, I think a greater debate between children of deaf parents and children of hearing parents is interesting…. CODAs don't even appear in the discussion, and it's very interesting.
|
As can be understood from the study title, this study was conducted to determine the experiences of hearing parents whose children have hearing loss. These experiences and feelings were discussed in the light of the literature. We believe that the discussion should not have mentioned CODA, as the study aimed not to compare parents' experiences with and without hearing loss. |

Reviewer 2 Report
The uthors answered my questions
Author Response
Reviewer II Comment
The authors answered my questions
Author Response
Thank you for your contribution. We believe that the manuscript was enhanced and improved after your recommendations.